# Initiation of domiciliary care and nursing home admission following first hospitalization for heart failure, stroke, chronic obstructive pulmonary disease or cancer

**Rasmus Rørth**[1]*, **Marianne F. Clausen**[1], **Emil L. Fosbøl**[1], **Ulrik M. Mogensen**[1], **Kristian Kragholm**[2], **Pardeep S. Jhund**[3], **Mark C. Petrie**[3], **Christian Torp-Pedersen**[4], **Gunnar H. Gislason**[5], **John J. V. McMurray**[3], **Lars Køber**[1], **Søren L. Kristensen**[1]

**1** Department of Cardiology, Rigshospitalet, University of Copenhagen, Copenhagen, Denmark,
**2** Department of Cardiology and Clinical Epidemiology, Aalborg University Hospital, Aalborg, Denmark,
**3** BHF Cardiovascular Research Centre, Institute of Cardiovascular and Medical Sciences, University of Glasgow, Glasgow, United Kingdom, **4** North Zealand University Hospital, Hillerød, Denmark, **5** Department of Cardiology, Gentofte/Herlev University Hospital, Copenhagen, Denmark

\* rasmusroerth@hotmail.com

**Data Availability Statement:** Data cannot be shared publicly because they include sensitive information. Data are available from Statistics

## Abstract

### Background

Patients with chronic diseases are at higher risk of requiring domiciliary and nursing home care, but how different chronic diseases compare in terms of risk is not known. We examined initiation of domiciliary care and nursing home admission among patients with heart failure (HF), stroke, COPD and cancer.

### Methods

Patients with a first-time hospitalization for HF, stroke, COPD or cancer from 2008–2016 were identified. Patients were matched on age and sex and followed for five years.

### Results

111,144 patients, 27,786 with each disease, were identified. The median age was 69 years and two thirds of the patients were men. The 5-year risk of receiving domiciliary care was; HF 20.9%, stroke 25.2%, COPD 24.6% and cancer 19.3%. The corresponding adjusted hazard ratios (HRs), with HF patients used as reference, were: stroke 1.35[1.30–1.40]; COPD 1.29[1.25–1.34]; and cancer 1.19[1.14–1.23]. The five-year incidence of nursing home admission was 6.6% for stroke, and substantially lower in patients with HF(2.6%), COPD(2.6%) and cancer (1.5%). The adjusted HRs were (HF reference): stroke, 2.44 [2.23–2.68]; COPD 1.01 [0.91–1.13] and cancer 0.76 [0.67–0.86]. Living alone, older age, diabetes, chronic kidney disease, depression and dementia predicted a higher likelihood of both types of care.

Denmark (Contact information: Email: dst@dst.dk; Telephone: +45 39 17 39 17) for researchers who meet the criteria for access to confidential data.

**Funding:** The authors received no specific funding for this work.

**Competing interests:** The authors have declared that no competing interests exist.

## Conclusions

In patients with HF, stroke, COPD or cancer 5-year risk of domiciliary care and nursing home admission, ranged from 19–25% and 1–7%, respectively. Patients with stroke had the highest rate of domiciliary care and were more than twice as likely to be admitted to a nursing home, compared to patients with the other conditions.

## Introduction

Patients with chronic diseases such as heart failure (HF), stroke, chronic obstructive pulmonary disease (COPD) and cancer are at a considerable risk of hospital admission and death but little is known about the consequence of these conditions on the ability of patients to live, unaided, in the community [1–5]. As a result of improvement in treatment, most patients with chronic diseases are living longer and getting older, underscoring the importance of looking beyond mortality and hospitalizations when assessing the impact of each of these conditions on patients and society. One approach would be to assess quality of life and functional status, as defined by capability to perform activities of daily living. However, it is difficult to measure quality of life in large-scale patient cohorts, and little is known how well these patients are living with their disease. Similarly, functional capacity is hard to measure in community studies, although these conditions may cause fatigue and impaired functional status, possibly preventing patients from carrying out normal activities of daily living and managing independently at home [6–9]. Domiciliary support such as help with shopping, meal preparation, personal care and cleaning and, in extreme cases, institutional care, for example nursing home admission, may be needed and can be used as objective measures of patients' autonomy and ability to live unaided. Also, domiciliary care and nursing home admission reflect an important measure of the personal, family and societal burden of chronic conditions. Loss of autonomy and possible separation from a spouse or family can lead to loss of self-esteem, indolence and depression. We know that patients with chronic conditions are at higher risk of needing these types of support than those without [10, 11]. But how the risk of domiciliary care and nursing home admissions compare between different types of chronic conditions is not known. To further evaluate these consequences of chronic diseases, we conducted a nationwide study in Denmark using cross-linkage of health and administrative registries.

## Methods

### Data sources

A unique personal identification number is assigned to all residents in Denmark which makes linkage of nationwide registries at an individual level possible [12]. Danish nationwide registries hold information on sociodemographic characteristics, all hospital admissions and claimed drug prescriptions [13, 14]. Data on domiciliary care in the community have been available since 2008 and data on nursing home admission since 1994 [15]. The study was approved by the Danish Data Protection Agency and data are available from Statistics Denmark upon application for researchers located in Denmark. Register-based studies in which individuals cannot be identified do not require ethical approval in Denmark.

### Study population and baseline variables

We designed a retrospective cohort study of all Danish residents with a first ever hospitalization for HF, stroke, COPD or cancer between January 1, 2008 and December 31, 2016 and followed these patients for up to five years. Patients were required to be alive at discharge and were excluded if they had received domiciliary care, were living in a nursing home, or had any of the other conditions of interest prior to study inclusion. Each patient entered the study on their date of discharge and was individually matched with patients from the other groups of diseases based on age, sex and calendar year. All matched patients were followed until an outcome of interest occurred, death, for a maximum of 5 years or until end of study (December 31, 2017). Comorbidities were identified by hospital discharge ICD-10 codes in a 10-year period prior to qualifying hospitalization; S1 Appendix. Diabetes mellitus, dementia and depression were additionally identified by at least one filled prescription for a disease-specific drug in the preceding 6 months. Ongoing use of medication was defined by at least one filled prescription of the drug in the preceding 6 months or 7 days after discharge but was not included in the adjusted Cox regression analyses; S2 Appendix.

### Outcome measures

The primary outcomes were initiation of domiciliary care and admission to a nursing home. Domiciliary care was defined as help given if there are tasks in the home that the citizen can no longer carry out themselves. In Denmark domiciliary care covers three main areas: 1) Personal care, including bathing, dressing and eating, 2) Practical help such as shopping, cleaning and doing laundry, and 3) Food service [16]. Nursing home is defined as an institution where citizens live if they can no longer take care of themselves.

### Statistics

Baseline characteristics for HF, stroke, COPD and cancer patients were described by use of proportions for categorical variables and medians/quartiles for continuous variables. Cumulative incidence curves for initiation of domiciliary care and nursing home admission, with death as a competing risk, were estimated using the Aalen-Johansen method and differences between the diseases were compared using Gray's test [17, 18]. We also used cause-specific Cox regression to compare the risk of initiation of domiciliary care and nursing home admission between patient groups. The Cox regression analyses were adjusted for age, sex, marital status, calendar year, and comorbidities (ischemic heart disease, atrial fibrillation, chronic kidney disease and diabetes) and stratified by who each patient was matched with. Adjusted variables were chosen before any analysis was done and were based on clinical relevance and known prognostic importance. The variables sex, age, calendar year and marital status were tested for interactions with the diseases in relation to both outcomes. Log (-log(survival)) curves were used to evaluate the proportional hazard assumption. The assumption of linearity of age was tested by including a variable of age squared.

The SAS statistical software package, version 9.4 (SAS Institute, Cary, North Carolina; USA) and and R, version 3.5.1 (R development Core Team) were used for all analyses.

## Results

### Baseline characteristics

Following matching on age, sex and calendar year we identified 111,144 patients, 27,786 with each disease. Baseline characteristics of patients in the four disease groups are shown in Table 1. The median age was 69 and two thirds of the patients were men. Patients with HF

**Table 1. Baseline characteristics a first hospitalization for HF, stroke, COPD or cancer.**

|  | Heart failure | Stroke | COPD | Cancer |
|---|---|---|---|---|
| **No. Patients** | 27786 | 27786 | 27786 | 27786 |
| Age, median (Q1-Q3) | 69 (60–76) | 69 (60–76) | 69 (60–76) | 69 (60–76) |
| Male | 18502 (67) | 18502 (67) | 18502 (67) | 18502 (67) |
| **Civil status (%)** |  |  |  |  |
| Living alone | 8902 (32) | 8636 (31) | 10155 (37) | 7412 (27) |
| **Comorbidities (%)** |  |  |  |  |
| IHD | 16339 (59) | 5593 (20) | 7059 (25) | 3956 (14) |
| Atrial fibrillation | 12663 (46) | 5570 (20) | 5372 (19) | 3091 (11) |
| Diabetes | 6967 (25) | 4598 (17) | 4514 (16) | 3387 (12) |
| CKD | 4464 (16) | 1559 (6) | 2173 (8) | 1428 (5) |
| Dementia | 195 (0.7) | 309 (1.1) | 216 (0.8) | 135 (0.5) |
| Depression | 3247 (12) | 4198 (15) | 4531 (16) | 2490 (9) |
| **Pharmacotherapy* (%)** |  |  |  |  |
| Antiplatelets, any | 15209 (55) | 18806 (68) | 7830 (28) | 5237 (19) |
| Lipid-lowering drugs | 14594 (52) | 17423 (63) | 8161 (29) | 6721 (24) |
| Thiazides | 4298 (15) | 4463 (16) | 4114 (15) | 3319 (12) |
| Loop diuretics | 16381 (59) | 1848 (7) | 5004 (18) | 1988 (7) |
| Beta blockers | 19668 (71) | 6320 (23) | 6110 (22) | 4350 (16) |
| ACE-I/ARB | 20761 (75) | 11461 (41) | 9149 (33) | 8098 (29) |

COPD—chronic obstructive pulmonary disease; IHD—ischemic heart disease;

ACE-I—angiotensin-converting enzyme inhibitors, ARB—angiotensin-II receptor blockers;

*Filled in prescriptions 180 days prior to admission or 7 days after discharge.

were more likely to have ischemic heart disease, atrial fibrillation, diabetes and chronic kidney disease that the others. Patients with stroke had the highest proportion with dementia and depression and patients with COPD were more likely to be living alone.

## Initiation of domiciliary care

Over five years of follow-up (median time: 1395 days; quartile 1- quartile 3: 670–1825 days) the need for domiciliary care was 20.9% [20.4%–21.4%] in HF patients, 25.2% [24.7%–25.7%] in stroke patients, 24.6% [24.1%–25.1%] in patients with COPD and 19.3% [18.8%–19.7%] in patients with cancer; P< 0.0001; Fig 1A. The competing risk of death was 16.2% [15.7%–16.6%], 10.0% [9.6%–10.3%], 17.3% [16.9%–17.8%] and 25.4% [24.9%–26.0%]; P<0.0001, among HF-, stroke-, COPD- and cancer patients respectively; Fig 1B. In adjusted analyses with HF patients as reference group, the likelihood of initiating domiciliary care was significantly higher for both stroke—(HR 1.35 [1.30–1.40]), COPD—(HR 1.29 [1.25–1.34]) and cancer patients (HR 1.19 [1.14–1.23]); Fig 2. Other factors associated with initiation of domiciliary support included older age (HR 1.06 [1.06–1.06] per 1-year increase in age), female gender (HR 1.14 [1.11–1.17]) and living alone (HR 1.92 [1.88–1.98]); Fig 2. Living alone was associated with a greater need for domiciliary care in both men than women; however, the association was significantly stronger in men (men: HR 2.22 [2.14–2.29] vs women: HR 1.60 [1.53–1.63]; P for interaction<0.0001). Most comorbidities were also associated with higher likelihood of domiciliary care especially dementia (HR 1.58 [1.44–1.75]), depression (HR 1.56 [1.51–1.62]) and chronic kidney disease (HR 1.53 [1.47–1.59]) but also diabetes (HR 1.26 [1.22–1.30]) and atrial fibrillation (HR 1.13 [1.10–1.18]).

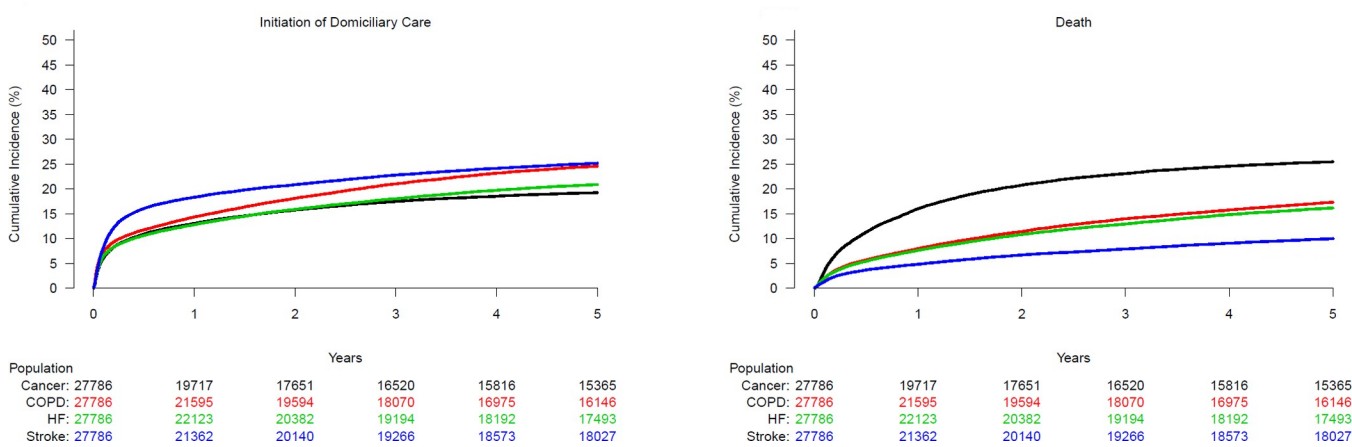

**Fig 1.** Cumulative incidence of domiciliary care initiation (A) with death as a competing risk (B) in patients with HF, stroke, COPD or cancer.

## Nursing home admission

The overall incidence of nursing home admission was low, but large differences were observed between groups; patients with stroke had an incidence of 6.6% [6.4%–6.9%] compared to

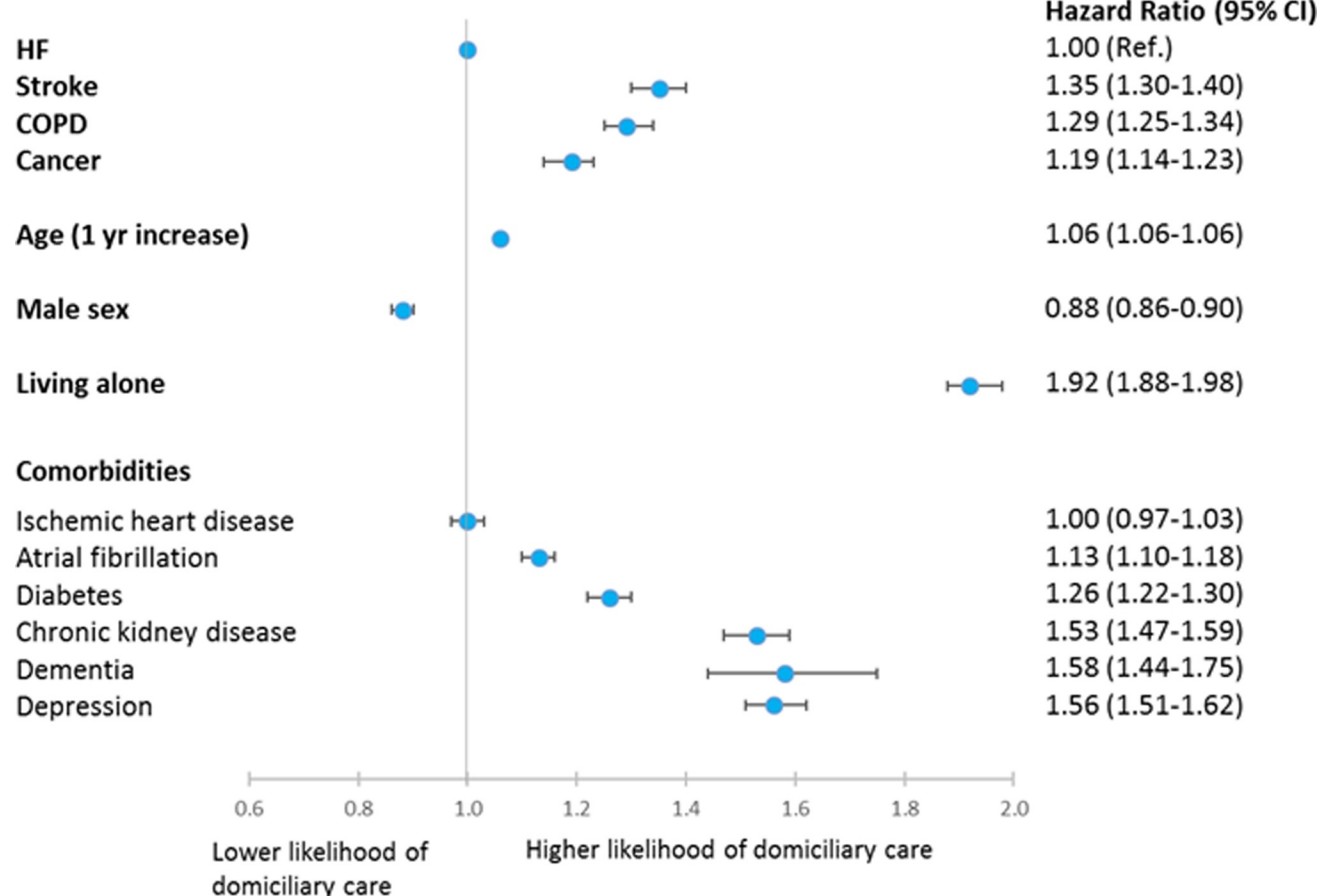

**Fig 2. Multivariable Cox regression model of factors associated with initiation of domiciliary care in patients with HF, stroke, COPD or cancer.**

patients with HF 2.6% [2.4%–2.8%], COPD 2.6% [2.5%–2.8%] and cancer 1.5% [1.4%–1.7%]; P<0.0001; Fig 3A. The competing risk of death was highest in patients with cancer (37.8% [37.2%–38.4%]), similar in patients with COPD 27.6% [27.1%–28.2%] and HF 24.4% [23.9%–24.9%], and lowest among patients with stroke 14.2% [13.8%–14.6%]; P<0.0001; Fig 3B. In adjusted analyses, with HF patients as reference, the likelihood of nursing home admission was more than twice as high for patients with stroke (HR 2.44 [2.23–2.68]), the same for patients with COPD (HR 1.01 [0.91–1.13]), and significantly lower for patients with cancer HR = 0.76 [0.67–0.86]; Fig 4. As for domiciliary care, age (HR 1.06 [1.06–1.06] per 1-year increase in age) and living alone (HR 1.92 [1.88–1.98]) were strongly associated with nursing home admission. The association of living alone and nursing home admission was seen in both men (HR 2.74 [2.52–2.98]) and women (HR 1.58 [1.42–1.77]); but was significantly stronger in men (P for interaction <0.001). In contrast to the findings regarding domiciliary care, male gender (HR 1.08 [1.01–1.16]) was associated with a higher likelihood of nursing home admission. As for comorbidities, dementia (HR 4.00 [3.47–4.61]) and depression (HR 2.14 [1.98–2.31]) were particularly associated with admission to a nursing home, as were chronic kidney disease (HR 1.21 [1.10–1.34]) and diabetes (HR 1.31 [1.20–1.41]) (Fig 4).

## Discussion

Domiciliary care was initiated in around 20–25% of patients within 5 years of discharge, following a first hospitalization for either HF, stroke, COPD or cancer. Patients with HF had a significant lower likelihood of needing this assistance than patients with stroke (who had the highest likelihood), COPD or cancer. Nursing home admission occurred in 2–7% of patients within 5 years, with the incidence more than twice as high in patients with stroke compared to the other diseases. Factors associated with both initiation of domiciliary care and nursing home admission included older age, living alone and non-cardiac comorbidities.

The conditions we examined, and other chronic diseases, are characterised by symptoms such as fatigue breathlessness and associated with reduced physical ability, all of which tend to worsen over time and may lead to loss of independence and the need for support in the community or residential care. Between 1 in 4 and 1 in 5 of patients in our study had domiciliary care initiated within 5 years of discharge, and most needed within 1 year. This highlights a different but important aspect of disease trajectory not necessary is reflected by hospitalizations or mortality and likely reflecting progressive worsening of symptoms and functional status.

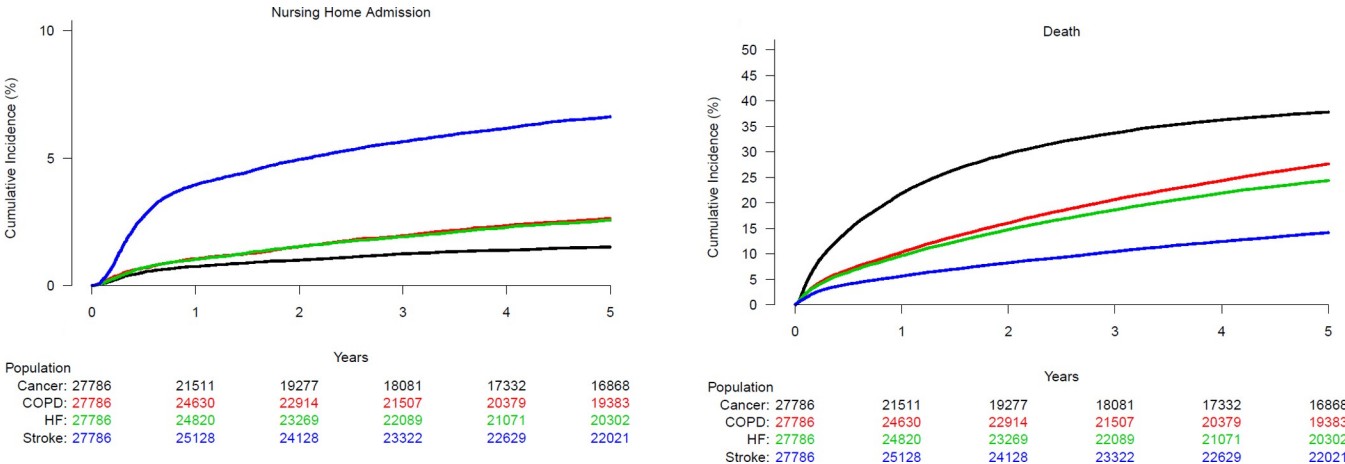

**Fig 3.** Cumulative incidence of nursing home admission (A) with death as a competing risk (B) in patients with HF, stroke, COPD or cancer.

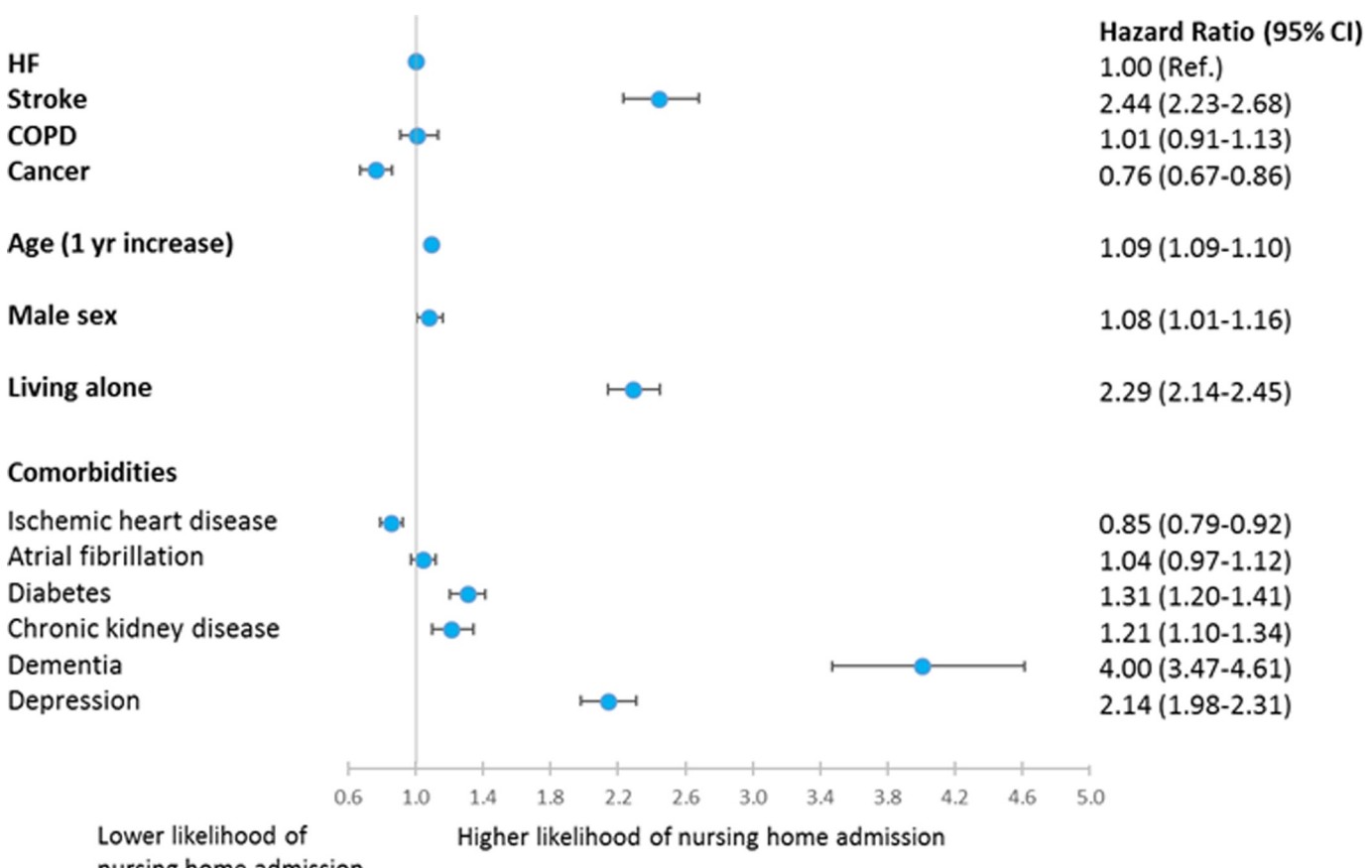

**Fig 4. Multivariable Cox regression model of factors associated with nursing home admission in patients with HF, stroke, COPD or cancer.**

Furthermore, it reflects an important impact of these chronic conditions at both the level of the individual patient and for society more generally. Thus, domiciliary and nursing home care reflect the broader societal burden of these chronic conditions and may be valuable additional metrics of the effect of treatment on chronic diseases, their economic consequences and, potentially, quality of care.

The need for domiciliary care and, especially, the incidence of nursing home admission, was highest among stroke patients. This may reflect the particular effect of stroke on vision, speech, swallowing, use of limbs, and bladder control all of which may make affected individuals dependent on the help of others [19]. HF, COPD and cancer don't usually cause this same range of disabilities and this may explain the higher need for community and residential care associated with stroke, compared with these other conditions. Although patients with stroke had a markedly higher risk of nursing home admission, it is important to point out that the absolute risk of nursing home admission was low. Furthermore, death was a major competing risk for the other conditions, especially in cancer, which might have led to an underestimation of the real differences in utilization of these support services.

It was somewhat surprising that both COPD and cancer patients were significantly more likely than patients with HF to have domiciliary care initiated after hospital discharge. For COPD this might be related to the condition being associated with lower social status and a higher proportion of patients with COPD living alone; this may mean that such patients had less personal support to allow them to live without help [20]. With respect to cancer, this

condition may still be perceived as a more deadly than HF, making access to benefits such as domiciliary care easier [21]. However, this is hypothesis is speculative and need further investigation and verification.

Other factors such as the impact of comorbidities, mental health, cognitive function and social support beyond severity of the chronic disease might be important in the evaluation of the need to initiate domiciliary care, or to admit a patient to a nursing home. Indeed, we saw a significant association between comorbidities such as diabetes and chronic kidney disease with both domiciliary care and nursing home admission. Of note, comorbidities which are linked to mental health problems and cognitive dysfunction, i.e. depression and dementia showed a particularly strong association with each outcome, indicating that mental health and cognitive function might be as important as physical limitations in determining whether independent living can be maintained. The importance of spousal support was evident from the significant association of living alone and likelihood of both domiciliary care and nursing home admission. Notably we found a significant interaction of sex and living alone on both outcomes, i.e. to be living alone had a higher impact on men than on women, presumably reflecting that women in general are better to take care of themselves.

## Strengths and limitations

The main strengths of our study are that we have a nationwide unselected cohort of patients with a first ever hospitalization for either HF, stroke, COPD or cancer followed in a real life setting with complete follow-up on all patients, except those who emigrated from Denmark during the study period. The main limitations are absence of certain clinical variables, such as HF severity; cancer type, stage and treatment; performance status, stroke score and pulmonary function tests which could help us identify the severity of the conditions of interest. Together with possible other unmeasured confounders these factors could have influenced our results in ways we have not been able to account for in our analyses. Thus, as is always the case with observational studies, the associations we found are not necessarily causal. Data on hours and type of domiciliary care are not adequately collected and therefore domiciliary care may cover a wide spectrum of services. We did not have access to data on self-care autonomy such as activities of daily living. Finally, our findings were based on the Danish healthcare and social systems and therefore may not be applicable to other countries.

## Conclusions

One out of five patients have domiciliary care initiated within 5 years of first hospitalization for HF, stroke, COPD or cancer. Patients with stroke were most likely to receive domiciliary care and were more than twice as likely to be admitted to a nursing home as patients with any of the other conditions examined. Comorbidities, living alone and older age were associated with higher likelihood of receiving each type of care. Domiciliary care and nursing home admission may be valuable additional metrics of the impact of chronic diseases on both patients and on society, the effect of treatment on these conditions and of the quality of care.

## Supporting information

**S1 Appendix. ICD-10 codes.**
(DOCX)

**S2 Appendix. ATC classification codes.**
(DOCX)

## Author Contributions

**Conceptualization:** Rasmus Rørth, John J. V. McMurray, Lars Køber, Søren L. Kristensen.

**Formal analysis:** Ulrik M. Mogensen, Søren L. Kristensen.

**Methodology:** Rasmus Rørth, Marianne F. Clausen, Ulrik M. Mogensen, Christian Torp-Pedersen, Lars Køber.

**Resources:** Christian Torp-Pedersen.

**Supervision:** Emil L. Fosbøl, John J. V. McMurray, Lars Køber, Søren L. Kristensen.

**Visualization:** John J. V. McMurray.

**Writing – original draft:** Rasmus Rørth.

**Writing – review & editing:** Marianne F. Clausen, Emil L. Fosbøl, Ulrik M. Mogensen, Kristian Kragholm, Pardeep S. Jhund, Mark C. Petrie, Christian Torp-Pedersen, Gunnar H. Gislason, John J. V. McMurray, Lars Køber, Søren L. Kristensen.

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
