## [Decision Letter · Decision Letter 0]

7 Apr 2021

PONE-D-21-02090

Initiation of domiciliary care and nursing home admission following first hospitalization for heart failure, stroke, chronic obstructive pulmonary disease or cancer

PLOS ONE

Dear Dr. Rørth,

Thank you for submitting your manuscript to PLOS ONE. After careful consideration, we feel that it has merit but does not fully meet PLOS ONE’s publication criteria as it currently stands. Therefore, we invite you to submit a revised version of the manuscript that addresses the points raised during the review process.

We look forward to receiving your revised manuscript.

Kind regards,

Antonio Cannatà

Academic Editor

PLOS ONE

Journal Requirements:

Reviewers' comments:

Reviewer's Responses to Questions

**Comments to the Author**

1. Is the manuscript technically sound, and do the data support the conclusions?

Reviewer #1: Yes

Reviewer #2: Yes

2. Has the statistical analysis been performed appropriately and rigorously? 

Reviewer #1: Yes

Reviewer #2: Yes

3. Have the authors made all data underlying the findings in their manuscript fully available?

Reviewer #1: No

Reviewer #2: Yes

4. Is the manuscript presented in an intelligible fashion and written in standard English?

Reviewer #1: Yes

Reviewer #2: Yes

5. Review Comments to the Author

Reviewer #1: Overall Comments:

Thank you for the opportunity to review this manuscript. This is an interesting retrospective cohort study using Danish nationwide registries to compare risk of needing domiciliary care or nursing home care between patients with HF, stroke, COPD, and cancer. The statistical methods that the authors employ appear to be appropriate, though I am not a statistician myself. I have a few comments/questions outlined below:

Methods

- In terms of clarity, I suggest the authors start with describing the overall approach to this study. It appears to be a retrospective cohort study.

- page 11 “ Comorbidities were identified by hospital discharge codes in a 10-year period prior to qualifying hospitalization”

o Please provide more details about this---do you mean ICD10 codes that are coded/associated with that particular admission?

- page 11 “Diabetes mellitus, dementia and depression were additionally identified by at least one filled prescription for a disease-specific drug in the preceding 6 months.”

o Please provide an Appendix where details on which disease-specific drugs you used to define these co-morbidities.

- page 11, “Ongoing use of medication was defined by at least one filled prescription of the drug in the preceding 6 months or 7 days after discharge but was not included in the adjusted Cox regression analyses.”

o Do you mean the variable reflecting filled prescription was not included in the model?

Results

- Would be interesting to know how many patients in each cohort (HF/stroke/COPD/cancer) developed one of the other conditions of interest over the follow up period, and if development of additional conditions of interest affected risk of receiving domiciliary care or admission to nursing home. Patients at particularly high risk for needing receiving domiciliary care or admission to nursing home likely also have multimorbidity of these illnesses.

- The rate of nursing home admission in this study is surprisingly very low. In the US it is much more common (https://www.hcup-us.ahrq.gov/reports/statbriefs/sb205-Hospital-Discharge-Postacute-Care.jsp ). This may be because the differences in how domiciliary care and nursing home care is funded between the two countries. In the US, it is cheaper for insurance companies to pay for nursing home care rather than a home health aid 24/7. This doesn’t necessarily reflect patient or family preferences. It could also potentially provide an example of what utilization of these services might be if our healthcare/society in the US was different. I know you briefly mention that your findings may not be applicable to other countries, but might be interesting to discuss what might be learned from the differences.

Reviewer #2: This is an interesting paper that describes the clinical management out of the hospital of patients affected by chronic diseases such as heart failure (HF), stroke, chronic obstructive pulmonary disease (COPD) and cancer.

I have some comments:

- It would be very interesting to evaluate the same outcomes in these two subgroups of patients: one of ‘’living alone’’ and other one ‘’supported by relatives’’. The family’s support could modify the necessity of domiciliary care initiation and nursing home admission.

- It would be very interesting a clinical evaluation of self-care autonomy of these patients at the baseline and during the follow-up, with the help of some specific scales such as ADLs or IADLs. If it is not applicable, please report it in the limitation.

- Further information about cancer disease of these patients are needed, such as the presence of metastasis, palliative or chemotherapy treatments. Moreover, at least NYHA class of the patients affected by HF should be reported, at the baseline and during follow-up.

- Did any of these patients need re-hospitalization during follow-up?

6. PLOS authors have the option to publish the peer review history of their article (what does this mean?). If published, this will include your full peer review and any attached files.

Reviewer #1: **Yes: **Himali Weerahandi

Reviewer #2: No

---

## [Author Response · Author response to Decision Letter 0]

8 Jun 2021

Reviewer #1, general comments:

Thank you for the opportunity to review this manuscript. This is an interesting retrospective cohort study using Danish nationwide registries to compare risk of needing domiciliary care or nursing home care between patients with HF, stroke, COPD, and cancer. The statistical methods that the authors employ appear to be appropriate, though I am not a statistician myself. I have a few comments/questions outlined below:

Thank you for this comment on our manuscript.

Reviewer #1, comment no 1:

- In terms of clarity, I suggest the authors start with describing the overall approach to this study. It appears to be a retrospective cohort study.

Thank you for this comment. We have tried to make the description of the study design more clear.

Changes made to the manuscript: Method section, (P 5, L11-13)

We designed a retrospective cohort study of all Danish residents with a first ever hospitalization for HF, stroke, COPD or cancer between January 1, 2008 and December 31, 2016, were identified and followed these patients for up to five years.

Reviewer #1, comment no 2:

- page 11 “ Comorbidities were identified by hospital discharge codes in a 10-year period prior to qualifying hospitalization”

o Please provide more details about this---do you mean ICD10 codes that are coded/associated with that particular admission?

Thank you for pointing this out. Yes, we used ICD codes from prior hospital admissions. A more detailed description and an appendix has now been included.

Changes made to the manuscript: Method section, (P 5, L19-20)

Comorbidities were identified by hospital discharge ICD-10 codes in a 10-year period prior to qualifying hospitalization; Appendix 1.

Appendix 1 ICD-10 codes

Comorbidity ICD-10 codes

Heart failure I420, I426, I427, I428, I429, I50, I110, I130, I132

Stroke I60-64

Chronic obstructive pulmonary disease J42-J44

Cancer C00-C97 (if not C44)

Ischemic heart disease I20-I25

Atrial fibrillation I48

Diabetes E10-14

Chronic kidney disease E102, E112, E132, E142, I120, N02-N08, N11, N12, N14, N18, N19, N26, N158-N160, N162-N164, N168, M300, M313, M319, M321B, Q612, Q613, Q615, Q619, T858, T859, Z992

Dementia F00-F03

Depression F31-34

ICD, International Classification of Diseases 

Reviewer #1, comment no 3:

- page 11 “Diabetes mellitus, dementia and depression were additionally identified by at least one filled prescription for a disease-specific drug in the preceding 6 months.”

o Please provide an Appendix where details on which disease-specific drugs you used to define these co-morbidities.

An appendix with ATC codes has now been included.

Appendix 2. ATC classification codes

Pharmacotherapy ATC codes

Glucose lowering drugs

 A10

Dementia N06D

Depression N06A

Antiplatelets

 B01AC06, N02BA01, B01AC04, B01AC22, B01AC24, B01AC07

Lipid-lowering drugs

 C10

Thiazides

 C03A, C07B, C07D, C09XA52, C03EA01

Loop diuretics

 C03C, C03EB01, C03EB02

Beta-blockers

 C07, C09BX

Renin-angiotensin-system inhibitors

 C09

ATC, Anatomical Therapeutic Chemical

Reviewer #1, comment no 4:

- page 11, “Ongoing use of medication was defined by at least one filled prescription of the drug in the preceding 6 months or 7 days after discharge but was not included in the adjusted Cox regression analyses.”

o Do you mean the variable reflecting filled prescription was not included in the model?

Thank for this comment. Yes, we chose not to include medication in our COX regression models. We believe that this is a very difficult issue as the information provided by medication use is complex. The information that a given medication is used indicates a need, but is also indicates ability to tolerate the medication and a clinical condition which has good compliance with medical advice.

Reviewer #1, comment no 5:

- Would be interesting to know how many patients in each cohort (HF/stroke/COPD/cancer) developed one of the other conditions of interest over the follow up period, and if development of additional conditions of interest affected risk of receiving domiciliary care or admission to nursing home. Patients at particularly high risk for needing receiving domiciliary care or admission to nursing home likely also have multimorbidity of these illnesses.

This is another very interesting question. However, this was the scope of this manuscript. The purpose of the manuscript was to provide the clinicians/physisians with risk estimates of nursing home admissions and need for domiciliary care when they sit in front of their patients and furthermore be able to put this risk into context with other chronic diseases.

Reviewer #1, comment no 6:

- The rate of nursing home admission in this study is surprisingly very low. In the US it is much more common (https://www.hcup-us.ahrq.gov/reports/statbriefs/sb205-Hospital-Discharge-Postacute-Care.jsp ). This may be because the differences in how domiciliary care and nursing home care is funded between the two countries. In the US, it is cheaper for insurance companies to pay for nursing home care rather than a home health aid 24/7. This doesn’t necessarily reflect patient or family preferences. It could also potentially provide an example of what utilization of these services might be if our healthcare/society in the US was different. I know you briefly mention that your findings may not be applicable to other countries, but might be interesting to discuss what might be learned from the differences.

This is a very good and important point which we have also mentioned in our limitation. We have now tried to elaborate on this point. 

Changes made to the manuscript: Limitation section, (P 10, L12-15)

The main limitations are absence of certain clinical variables, such as HF severity; cancer type, stage and treatment; performance status, stroke score and pulmonary function tests which could help us identify the severity of the conditions of interest.

Reviewer #2, general comments

 This is an interesting paper that describes the clinical management out of the hospital of patients affected by chronic diseases such as heart failure (HF), stroke, chronic obstructive pulmonary disease (COPD) and cancer.

Thank you.

Reviewer #2, comment no 1:

- It would be very interesting to evaluate the same outcomes in these two subgroups of patients: one of ‘’living alone’’ and other one ‘’supported by relatives’’. The family’s support could modify the necessity of domiciliary care initiation and nursing home admission.

The reviewer raises some very interesting points. 

Living alone was included in all our model and shown to be a strong predictor of both the need for domiciliary care and nursinghome admissions. Further, there was no interaction between living alone and the respective chronic diseases.

Unfortunately, we don’t have data on support from relatives.

Reviewer #2, comment no 2:

- It would be very interesting a clinical evaluation of self-care autonomy of these patients at the baseline and during the follow-up, with the help of some specific scales such as ADLs or IADLs. If it is not applicable, please report it in the limitation.

We agree that information such as ADL could provide valuable extra information. Unfortunately we don’t have date on self-ca autonomy. We have now included this in the limitation section.

Changes made to the manuscript: Limitation section, (P 10, L19)

We did not have access to data on self-care autonomy such as activities of daily living

Reviewer #2, comment no 3:

- Further information about cancer disease of these patients are needed, such as the presence of metastasis, palliative or chemotherapy treatments. Moreover, at least NYHA class of the patients affected by HF should be reported, at the baseline and during follow-up.

We agree that information such as NYHA class, ejection fraction, performance status and other measures of disease severity could provide valuable information. Unfortunately, we don’t have information regarding this which we now have highlighted in the limitation section.

Changes made to the manuscript: Limitation section, (P 10, L 19-21)

Finally, our findings were based on the Danish healthcare and social systems and therefore may not be applicable to other countries.

Reviewer #2, comment no 4:

- Did any of these patients need re-hospitalization during follow-up?

This is another interesting question, which was not the scope of the manuscript.

---

## [Decision Letter · Decision Letter 1]

15 Jul 2021

Initiation of domiciliary care and nursing home admission following first hospitalization for heart failure, stroke, chronic obstructive pulmonary disease or cancer

PONE-D-21-02090R1

Dear Dr. Rørth,

We’re pleased to inform you that your manuscript has been judged scientifically suitable for publication and will be formally accepted for publication once it meets all outstanding technical requirements.

Kind regards,

Antonio Cannatà

Academic Editor

PLOS ONE

Additional Editor Comments (optional):

Reviewers' comments:

Reviewer's Responses to Questions

**Comments to the Author**

1. If the authors have adequately addressed your comments raised in a previous round of review and you feel that this manuscript is now acceptable for publication, you may indicate that here to bypass the “Comments to the Author” section, enter your conflict of interest statement in the “Confidential to Editor” section, and submit your "Accept" recommendation.

Reviewer #1: All comments have been addressed

2. Is the manuscript technically sound, and do the data support the conclusions?

Reviewer #1: Yes

3. Has the statistical analysis been performed appropriately and rigorously? 

Reviewer #1: Yes

4. Have the authors made all data underlying the findings in their manuscript fully available?

Reviewer #1: No

5. Is the manuscript presented in an intelligible fashion and written in standard English?

Reviewer #1: Yes

6. Review Comments to the Author

Reviewer #1: I do still think the discussion should be expanded as I recommended in comment 6. Otherwise, the authors' responses are adequate.

7. PLOS authors have the option to publish the peer review history of their article (what does this mean?). If published, this will include your full peer review and any attached files.

Reviewer #1: **Yes: **Himali Weerahandi

---

## [Editor Report · Acceptance letter]

26 Jul 2021

PONE-D-21-02090R1 

Initiation of domiciliary care and nursing home admission following first hospitalization for heart failure, stroke, chronic obstructive pulmonary disease or cancer 

Dear Dr. Rørth:

I'm pleased to inform you that your manuscript has been deemed suitable for publication in PLOS ONE. Congratulations! Your manuscript is now with our production department. 

Kind regards, 

on behalf of

Dr. Antonio Cannatà 

Academic Editor

PLOS ONE